# Insight into the Structural Basis for Dual Nucleic Acid—Recognition by the Scaffold Attachment Factor B2 Protein

**DOI:** 10.3390/ijms24043286

**Published:** 2023-02-07

**Authors:** Sophie M. Korn, Julian Von Ehr, Karthikeyan Dhamotharan, Jan-Niklas Tants, Rupert Abele, Andreas Schlundt

**Affiliations:** 1Institute for Molecular Biosciences, Biomolecular Resonance Center (BMRZ), Goethe University Frankfurt, Max-von-Laue-Str. 7-9, 60438 Frankfurt, Germany; 2IMPRS on Cellular Biophysics, Max-von-Laue-Str. 7-9, 60438 Frankfurt, Germany; 3Institute for Biochemistry, Goethe University Frankfurt, Max-von-Laue-Str. 9, 60438 Frankfurt, Germany

**Keywords:** scaffold attachment factor proteins, nuclear matrix, nuclear magnetic resonance spectroscopy, SAP domain, RRM domain, dual nucleic acid binding, chromatin, RNA processing, protein dynamics

## Abstract

The family of scaffold attachment factor B (SAFB) proteins comprises three members and was first identified as binders of the nuclear matrix/scaffold. Over the past two decades, SAFBs were shown to act in DNA repair, mRNA/(l)ncRNA processing and as part of protein complexes with chromatin-modifying enzymes. SAFB proteins are approximately 100 kDa-sized dual nucleic acid-binding proteins with dedicated domains in an otherwise largely unstructured context, but whether and how they discriminate DNA and RNA binding has remained enigmatic. We here provide the SAFB2 DNA- and RNA-binding SAP and RRM domains in their functional boundaries and use solution NMR spectroscopy to ascribe DNA- and RNA-binding functions. We give insight into their target nucleic acid preferences and map the interfaces with respective nucleic acids on sparse data-derived SAP and RRM domain structures. Further, we provide evidence that the SAP domain exhibits intra-domain dynamics and a potential tendency to dimerize, which may expand its specifically targeted DNA sequence range. Our data provide a first molecular basis of and a starting point towards deciphering DNA- and RNA-binding functions of SAFB2 on the molecular level and serve a basis for understanding its localization to specific regions of chromatin and its involvement in the processing of specific RNA species.

## 1. Introduction

Recent studies have revealed a large number of canonical RNA-binding proteins (RBPs), now known to be involved in chromatin remodeling and transcriptional regulation [1]. Similarly, cell-wide proteomics-based studies have suggested that numerous DNA-binding proteins (DBPs) carry out functional roles in the processing of RNAs. In a rough classification, DRBPs can perform mutually exclusive or simultaneous DNA/RNA binding [2]. Interestingly, they can act as binders of the same gene on the DNA and transcript level, e.g., known for SMAD4 [3] and NF90 [4,5]. While for few proteins, a dual nucleic acid-binding capability using identical domains was proven on the structural level [6,7,8,9], we have only limited insight into how simultaneous DNA/RNA binding is achieved.

A canonical class of DRBPs is represented by the group of scaffold attachment factor (SAF) proteins. Type A SAF (SAFA, SAF-A), also known as hnRNP U, had been identified as an RNA binder and interactor with chromatin in the early 1990s almost simultaneously [10,11]. The family of type B SAF proteins (SAFB) comprises three members: SAFB1, SAFB2 and the SAFB-like transcriptional modulator SLTM. Similar to SAFA, they were first identified as binders of the nuclear matrix [12], where they interact with AT-rich DNA within scaffold/matrix attachment regions (S/MARs). Very early, SAFBs were also found to locate to subnuclear structures together with the protein HSF1 upon heat stress [13], which is in line with recent findings of SAFBs in the context of heat stress recovery [14]. Meanwhile, SAFBs are shown to act in DNA repair, transcriptional control, mRNA/(l)ncRNA processing, splicing regulation and as part of complexes with chromatin-modifying enzymes [15,16,17,18,19,20,21,22]. Particularly, SAFB2 was described as a factor to assist the processing of suboptimal stem–loop structures in miRNA maturation [23], while it abundantly locates to promoter regions and, e.g., interacts with the transcriptional regulator hnRNPUL1 [23]. Very recently, it was described in direct interactions with class-I transposable elements, where it apparently fulfills a function in their methylation [24]. 

SAFBs have been correlated with anti-tumor activity based on transcriptional repression [19,25], in particular their inhibitory interaction with transcription factors and nuclear receptors [22,26,27]. Human SAFB proteins are between 915 and 1034 amino acids long, and they all comprise highly-conserved SAP and RRM domains, primarily known for DNA and RNA binding, respectively, in their N-terminal half (Figure 1A–C). In addition, they contain a coiled coil (CC) and the little-investigated intrinsically disordered Glu-/Arg-rich (RE), Arg-/Gly-rich (RG/RGG) and Gly-rich regions (IDRs) in their C-terminal half. Those are primarily thought to confer protein–protein interactions (PPI), e.g., in oligomerization, interaction with DROSHA and SR proteins [15]. The CC region comprises a central role in PPIs of SAFBs with the protein ERH (enhancer of rudimentary homology) [15], which has a redundant function with SAFB2 [28]. The RGG part—based on its close relation with SAFA [10]—possibly functions in RNA engagement. For the RE stretch, li-terature suggests a role in the formation of nuclear stress bodies, foci of phase separation [29], in which SAFBs stabilize heterochromatin via interaction with transcripts [17]. Collectively, these functions involve the interaction with DNA and/or RNAs and, thus, require the presence of the dedicated nucleic acid-binding domains (NBDs) to confer specificity. 

Precise target sequences of SAFB NBDs have remained unknown, and so have a clear proof for the unambiguous NA-type preference for either of them. The ENCODE database (https://www.encodeproject.org/, accessed on 27 September 2022) lists a C(A/C)CCc motif derived from an RNA Bind-n-Seq (RBnS) experiment for full-length SAFB2 (identifier ENCSR558RBK) [33,34]. Such a C-rich short motif may well be bound by the RRM domain [35], specialized in ssRNA recognition in literally all RNA processes organized by RBPs [36]. The SAP domains, in turn, have an important role in maintaining chromatin structure, while their concrete target sequences are unknown. For SAFB2, there is no particular indication of a consensus target motif apart from its early categorization as an AT-rich DNA binder [12]. Other than for SAFB2, available ChIP-seq data [37] in the ENCODE database (identifier ENCSR072VUO) and follow-up studies have revealed precise binding sites with DNA promoter regions for SAFB1 [21], but a consensus motif was not derived. Of importance, SAFBs were shown to multimerize [23], presumably and with functional relevance, through their CC regions [23], which may support simultaneous functions and modulate target NA specificity through avidity. 

As of now, we lack any experimental structural information from all three SAFBs, i.e., their atom-resolved basis for specific interaction with DNAs, RNAs and proteins. Attempts toward structural insight into the dual nucleic acid binding of SAFB2, i.e., for example, RNA processing and the simultaneous anchoring to the nuclear matrix/transcription machinery interacting with DNA, have not been undertaken to date. 

We here set out to examine the basis of dual nucleic acid recognition by human SAFB2 via the dedicated NBDs. We used NMR spectroscopy to identify domain boundaries of its SAP and RRM domains and their stoichiometries and monitored their preferences for DNA and RNA motifs. Comprehensive NMR assignments facilitate mapping of binding sites with NAs to high-confidence structural models derived from sparse data and confirmed by mutations. Surprisingly, NMR data and additional biophysical experiments suggest the SAP domain to exhibit intra-domain dynamics and possibly to occur in a minor dimer conformation. Altogether, our findings represent a first important building block towards a comprehensive functional picture of SAFB D/RNPs suggested by their plastic compositions. 

## 2. Results

### 2.1. Definition of SAFB2 SAP Domain

SAP domains are described as 35-residue helix–loop–helix DNA-binding motifs (Figure 1B) [38]. The canonical SAP domain can be extended by flanking regions, contributing to affinity and/or specificity towards DNA targets [39]. Here, we cloned and expressed a 5.87 kDa region (residues 21 to 70) of human SAFB2 expected to comprise the canonical SAP domain and potential extensions (Appendix A). The quality of purified SAP was confirmed by Nuclear Magnetic Resonance (NMR) spectroscopy. The ^1^H-^15^N HSQC spectrum shows a well-dispersed peak pattern (Figure 2A), suggesting a folded protein species. This allowed a straightforward and complete backbone assignment of the natural sequence (Gly21 to Asp70) and the three artificial N-terminal residues (numbered Gly18-Ala19-Met20). As an exception, the amide of (non-native) Gly18 is line-broadened, caused by exchange with the solvent. Additionally, we were able to assign the sidechain amides of Asn46, Asn52 and Gln69. All backbone carbons were assigned except for the Gly18 Cα. Based on Cα/Cβ resonances, we calculated carbon secondary chemical shifts (SCS) of SAFB2 SAP, relative to random coil values (Figure 2B) [40]. Three consecutive residues with significant positive shifts (i.e., >1) were the basis for defining α-helices [41,42], for which we found the expected helix–loop–helix motif plus an additional one N-terminal to it. Thus, the SCS revealed SAFB2 SAP to constitute the canonical SAP fold. Interestingly, the SCS analysis also suggested the two inter-helical stretches to adopt medium and low β-strand propensity, respectively (defined as four consecutive amides with significant negative shifts, i.e., <1). We next measured the steady-state NOEs of NH resonances to visualize internal ns-ps dynamics within the SAP domain (Figure 2B). N- (18 to 29) and C-terminal residues (68 to 70) of the protein displayed high flexibility in solution, as indicated by hetNOE values below 0.5. This argues for a correctly defined length of the initial SAP domain construct suggested in this work. Residues 30 to 67 display hetNOE values above 0.65, with the exception of the stretch from 34 to 36. The increased flexibility of this stretch refutes a β-strand character, which had been indicated by the respective SCS values. Further, the loop connecting the two helices of the HLH motif (residues 46 to 51) exhibits rigidity—as evident from the high hetNOE values—in line with its subordinate β-strand propensity.

### 2.2. Definition of the SAFB2 RRM Domain Boundaries

A canonical RRM domain contains two conserved RNA-binding motifs (RNP1 and RNP2), embedded within a β-sheet surface (Figure 1C). Each motif is typically composed of aromatic residues that mediate base-stacking with target RNA molecules. The SAFB2 RRM domain was designed to comprise residues Gly404 to Asn485, in analogy to the human RBM5 RRM domain (PDB 2LKZ, [43]). The quality of the purified 9.13 kDa RRM domain (Appendix A) was confirmed by NMR spectroscopy. Like for SAP, RRM peaks in the ^1^H-^15^N HSQC spectrum were well-dispersed (Figure 3A). The backbone assignment is complete for all the 81 non-proline residues’ amides in the natural sequence, as well as two of the artificial N-terminal residues (numbered Gly401-Ala402-Met403 for convenience). Consequently, backbone carbons were assigned to completeness, and we also assigned sidechain amides of Asn408, Trp410, Asn425, Asn441 and Asn485 (Appendix A). SCS of SAFB2 RRM revealed the typical RRM fold: βα ββα ββ (Figure 3B) [35]. The {^1^H}^15^N heteronuclear NOE spectra revealed the RRM termini to be flexible and, thus, the domain boundaries not to disrupt the structure. A flexible loop between β2 and β3 confirmed the RRM to adopt the canonical RRM fold. 

### 2.3. Sparse NMR Data-Derived Structural Models of SAP and RRM Domains

NMR chemical shifts carry valuable structural information about the containing protein domains [44]. To gain deeper structural insight in the two SAFB2 domains, we next used domains as predicted by AlphaFold [45] and compared them to NMR-derived structures based on sparse data (i.e., the obtained assignments of chemical shifts as given in Section 2.1 and Section 2.2). We used the CS-Rosetta algorithm [46] and obtained well-converging ensembles for both the RRM and the SAP domains (Appendix A). Of note, both domains are in very good agreement with the models predicted by AlphaFold (Appendix A). The RRM forms a canonical domain-type following the β-α-β-β-α-β-β domain architecture. In support of Figure 3, the folded region comprises residues Gly408 to Lys482, and a comparison with selected RRMs derived from a DALI search [47] confirms the canonical nature of the SAFB2 RRM (Appendix A) [48].

Both AlphaFold and CS-Rosetta reveal a compact fold of SAP for the region between Arg29 and Gly68 (Appendix A), which is line with hetNOE data in Figure 2. The SAP domain reveals the anticipated helix–loop–helix core, a highly conserved structural motif within SAP domains, and fully converging for SAFB2 (Appendix A). Importantly, this motif is present in literally all types of SAP domains, irrespective of their functional context and DNA preferences (Appendix A) as suggested by the DALI-based structure alignment with the first three selected SAP domains according to the internal score [39,49,50]. Noteworthy, both the SAFB2 SAP structure suggested by CS-Rosetta as well as the AlphaFold model show a short N-terminal helical extension (Arg29-Leu33), which suggests a more occluded hydrophobic core of the domain. Apart from that, the region N-terminal to residue 29 is clearly unstructured and a truncated construct (26–70) reveals the same core domain fold compared to the main construct (21–70) as evidenced by comparison of HSQC spectra (Appendix A). In contrast, the C-terminal boundary, as used in our construct, is suggested to be embedded in a yet-structured context despite the lack of clear secondary structure beyond residue Gly68. 

### 2.4. Target RNA Preferences of RRM and Mapping of Binding Site

In order to determine target RNA preferences for the SAFB2 RRM, we compared CSPs induced by various RNAs (Table 1). Besides a set of Poly-N 7-mer RNAs (U_7_, A_7_, C_7_ and G_7_) we included a 6-mer RNA with the sequence 5′-UACACC-3′ (termed En+), based on the available ENCODE entry (identifier ENCSR558RBK) [33]. The RBnS - derived motif was obtained for pulldown against fl-SAFB2 and might reflect the RNA preferences of its RBD. As a control for the ENCODE motif, a 6-mer with a purine-core motif—5′-CGGACU-3′ (termed En−)—was tested.

In line with RNA-binding preferences described for a number of RRMs [51,52], we found C_7_ and A_7_ to induce significant CSPs of RNP residues (Figure 4A). U_7_ was the least-preferred motif. G_7_ could not be analyzed quantitatively, due to severe precipitation upon addition to RNA, likely induced by G-quadruplex formation. Interestingly, the RBNS-derived En+ sequence as well as the En− control resulted in similar, and in sum lower CSPs compared to A_7_ and C_7_ (Figure 4B). This, however, does not exclude a relevance of this motif in a genomic context. Of note, trajectories of CSPs were the same for all tested RNAs and only the maximum values varied in dependence of the RNA. This clearly argues for a shared interaction mechanism, but distinct affinities. 

We next aimed at determining the binding sites of respective nucleic acids to the RRM domain. To this end, we quantified CSPs and mapped significantly affected residues on the lowest-energy CS-Rosetta structure of the RRM domain (Figure 4C). As expected, we found the most perturbed residues to form the two classical RNP motifs of the domain, as well as two C-terminal residues extending the β-sheet surface. This indicates the canonical RRM interaction with RNA.

### 2.5. Target DNA-Preferences of SAP

We next sought to determine the DNA-binding capability of the SAFB2 putative DBD, SAP. EMSAs with fluorescently labelled 13-mer DNAs (Table 1), either GC- or AT-rich, reveal SAP’s slight preference for an AT over GC-rich content (Figure 5A). More pronounced, the SAFB2 SAP domain exclusively binds double-stranded DNA, clearly distinguishing it from other described SAP domains [39]. Since SAP’s apparent micromolar DNA-binding affinity makes it well suitable for NMR-analysis, we next aimed at atom-resolved insights into the SAP–DNA interaction. Stepwise titration of AT-rich dsAT_16mer_ up to 6-fold molar excess confirmed SAP binding to occur in a fast-exchange regime—indicative of micromolar affinity—for the majority of peaks (Figure 5B). We then quantified CSPs and mapped significantly affected residues on the lowest-energy CS-Rosetta structure of the SAP domain (Figure 5C,D). As expected, we found only few, but yet-more-significantly perturbed residues upon interaction with the AT-rich dsAT_16mer_ DNA. The same residues were perturbed upon interaction with GC-rich dsGC_16mer_, showing identical trajectories but overall lower maximum CSP values (Appendix A). No interaction was observed with ssAT_16mer_, in line with the suggested preferences from the EMSAs. Furthermore, reducing the DNA length to nine base pairs did not alter CSP values compared to the 16-mer, indicating dsAT_9mer_ to be sufficient for interaction with the SAP interface (Appendix A). 

Interestingly, three of the significantly perturbed residues cluster on one side of the domain and include the two highly conserved residues Lys53 and Ser54, plus an additional serine (Ser31) located in the N-terminal helical extension (Figure 1B). A fourth strongly affected residue, Arg45, was found on the opposite side of the domain. This suggests either a complex type of interaction between SAP and DNA, an indirect effect derived from intra-domain rearrangements upon DNA binding or a significant change in chemical environment based on SAP inter-domain interactions.

### 2.6. The SAFB2 RRM Domain Binds Both NA Types

For both NBDs of SAFB2, we found moderate, i.e., micromolar affinities with the expected nucleic acid type—RNA for the RRM and DNA for the SAP domain. Since many RRM domains are well described to also bind DNA, we next wondered about potential cross-interactions of the SAFB2 RRM and SAP domains with the non-dedicated nucleic acid, respectively. 

We first used a bona fide 34 nt model RNA (‘Bulge’, Table 1) with a well-distributed sequence composition comprising linear as well as structured regions [53]. A two-fold excess of Bulge-RNA resulted in significant CSPs of the RRM domain (Appendix A). In contrast, SAP does not interact with Bulge-RNA, as indicated by the lack of CSPs (Appendix A). Together with the results shown in Figure 5, this specifies SAFB2 SAP as an obligate dsDBD.

Vice versa, we then analyzed the RRM interacting with DNA. To that end, we used single- or double-stranded 16-mer DNAs, either AT- or GC-rich (Table 1, dsAT_16mer_, ssAT_16mer_ and dsGC_16mer_). Interestingly, the obtained CSP patterns indicate the RRM binds those DNAs with an equal extent and pattern of CSP compared to the tested poly-N RNAs (A_7_ and C_7_) in chapter 2.4 (Appendix A) pointing at an identical interaction surface. Similarly, the RRM seemingly exhibits an identical binding mode for both NA types, as suggested by the same CSP trajectories. The dual binding potential of SAFB2’s RRM domain might pose a particular and unique mechanistic feature.

### 2.7. SAP Domain Mutational Study

Based on our titration of SAP with the AT_16mer_ dsDNA (Figure 5) we next examined the contribution of single amino acids within the SAP domain to DNA binding. Therefore, we created mutants based on the CSP plot (Figure 5C) where we mutated the four amino acids with the highest CSPs to an alanine (S31A, R45A, K53A/S54A). The two neighboring amino acids K53/S54 were combined in one mutant. The influence of these amino acid exchanges on the domain fold was investigated by comparing ^1^H-^15^N HSQC spectra of the mutant SAP domains with the SAP_WT_ spectrum, respectively (Appendix A). Clearly, for SAP_S31A_ and SAP_K53A/S54A_ mutants, the folded state of the SAP domain is not impaired. Surprisingly, this was different for SAP_R45A_, which lost secondary structure based on the confined distribution of peaks in the HSQC, indicating a majorly unfolded species. 

We next analyzed the mutants’ abilities to bind dsDNA. Therefore, we recorded ^1^H-^15^N HSQCs at 6- (SAP_R45A_ and SAP_K53A/S54A_) or 8-fold (SAP_S31A_) molar excess of AT_9mer_ and compared them to their respective apo spectra (Figure 6). The lack of significant CSPs for SAP_K53AS54A_ suggests both residues to be decisive for DNA interaction. This is well in line with the literature, proposing the SAP–DNA interface to comprise the loop and the tips of both helices of the helix–loop–helix [39,54,55]. In contrast to that, SAP_S31A_ shows CSPs with trajectories comparable to the WT (Figure 6 and Appendix A) but with a lower total extent, which indicates that Ser31 increases SAP’s DNA-binding affinity but is not inevitably crucial for interaction. Lastly, SAP_R45A_ shows minor CSPs, which suggests some interaction of the protein with DNA. However, due to the loss of structure in the mutant, the protein–DNA interaction is most likely non-specific and rather based on the re-exposed positive charges of SAP residues engaging with the negatively charged DNA backbone. In support of NMR titrations, EMSAs with the fluorescently labelled 13-mer DNA (AT-rich) were found to confirm these results (Appendix A).

### 2.8. SAP Domain Intra- and Inter-Domain Dynamics

SAFB2 has been shown to form dimers, if not higher oligomers, including heteromultimers, with the other SAFB proteins [16,56,57]. While earlier work suggested the prominent CC region to convey protein–protein interactions, there is, to date, no full confinement of sequence stretches with regard to this. In fact, nucleic acid-binding specificity and affinity might be fostered by the coordinated recognition of adjacent motifs, i.e., dimers/oligomers of folded domains with or without DNA/RNA. RRMs are known to exist in a broad bandwidth of facets including the capability of mediating protein interactions. We, thus, first examined if the SAFB2 RRM domain exists in a monomeric form and used NMR ^15^N relaxation experiments to determine the total correlation time, a strong measure for the underlying molecular weight. We found a homogenous relaxation behavior of folded RRM residues supporting the compact domain. The average correlation time of 5.3 ns is clearly in accordance with a monomeric protein (Appendix A). 

For the SAP domain, we first performed in-detail inspection of their apparent oligomeric states via elution volumes in analytical size exclusion chromatography (aSEC) (Figure 7A). Surprisingly, molecular weight estimation based on calibration of an aSEC column with standards suggests that both the wild-type and the mutant SAP versions do not run as monomeric species (Figure 7B and Appendix A), but with MWs between monomers and dimers. This observation could indicate either protein oligomeric mixes or non-canonical retention by the column based on non-globular shapes, e.g., loss of compactness by intra-domain movements between the helices or even partial unfolding. The latter is best seen by the broad elution volume of SAP_R45A_ is in line with the loss of fold observed in the ^1^H-^15^N HSQC (Appendix A) and revealing an apparent MW larger than a dimer in aSEC. Nevertheless, this phenomenon is also seen for the N-terminally truncated SAP (26–70), suggesting that the diminished retention is caused by the core SAP fold, likely the HLH motif itself.

To obtain further insight into SAP oligomeric and dynamic features, we recorded ^15^N relaxation data for wild-type SAP (21–70). Surprisingly, individual amides show a large spread in R2 values leading to a large standard deviation (>50%). Motions on a µs-ms timescale can affect chemical exchange and thereby influence R2 rates. We, thus, measured R1rho relaxation, which allows to partially clear R2 from exchange contributions. Interestingly, we obtained an average total correlation time of 6 ns (Appendix A), estimating an apparent MW of 10 kDa in contrast to the theoretical 5.7 kDa, which represents a domain dimer, rather than a monomer. However, the individual amides still show a large spread in R1rho rates (Appendix A). We, thus, assume that the monomeric domain undergoes partial dimerization and experiences dynamics in form of exchange phenomena leading to the large variance in R2 rates. Of note, the obtained R1rho data for SAP WT yielded consistently reduced transversal relaxation rates, but a basically identical pattern throughout the domain, suggesting that SAP exists in dynamic equilibrium between conformers and /or a monomer–dimer mix. 

To finally obtain experimental molecular weights, we subjected all SAP version samples to SEC-MALS. As a result, the identified molar masses were in good agreement with monomers for all versions of SAP (Figure 7C), but all MWs are slightly higher than the theoretical ones (Appendix A). We, thus, conclude that the elution volumes observed in aSEC runs reflect both a mixture of monomer and dimer as well as intra-domain breathing, and SEC-MALS-derived MWs are that of a monomer with the influence of a low percentage of dimer in exchange. The latter is in line with the suggested intra- and inter-domain exchange observed via NMR relaxation experiments (Appendix A).

## 3. Discussion

SAFB proteins have been known for more than three decades and found to be involved in a large variety of nucleic acid-related biological functions [16], specifically in the context of stress response. Their supposedly dual nucleic acid-binding potential has made them bona fide anchoring proteins to locate and regulate DNA- and RNA-processing events, best represented by SAFB2 [23,58]. How exactly this is organized through SAFBs as integrators of RNA, DNA and protein binding has majorly remained elusive, while atom-resolved pictures would yield valuable insights into the respective functions. Surprisingly, for none of the three members has experimental structural information been publicly available. 

In this work, we provide the first atom-resolved analysis of the SAFB2 SAP and RRM domains and their dedicated interactions with DNA and RNA, respectively, as summarized graphically in Figure 8. Enabled by the suitable size of the two domains, we used NMR to unravel residue-resolved information of nucleic-acid binding and provide chemical shift-derived structural models. NMR is ideally applicable in this early step of a systematic deciphering of structure-guided contributions of the multi-domain SAFB2 to its integrated functions [59,60]. It was here used to optimize domain boundaries, determine the fold and oligomer states of domains, map their nucleic acid binding sites and report on internal dynamic regions.

For the SAFB2 RRM domain, we were able to obtain a complete protein backbone NMR assignment at pH 6.5. The few missing residues at pH 5.0 (see BMRB entry 51701) cluster in an exposed region of the compact domain, and, most likely, line-broadening under the acidic experimental conditions is caused by exchanging conformations. Interestingly, we find lysine 470, located centrally in this cluster, strongly affected by RNA binding. This suggests the local flexibility to be a determinant for the effective engagement with target sequences, and the entropic barrier of conformational selection acts as a proofread for specificity. The SAFB2 RRM, based on its preliminary sparse data-derived structure, appears to comprise a canonical fold and compares well to the majority of RRM structures. Thus, our identification of the binding interface with the tested RNAs collectively suggests an interaction via the RNP motifs, supported by possible contacts via flanking residues as, e.g., K470. Yet, we cannot rule out additional flanking regions or extensions [61] to modulate fold and RNA-motif recognition [35] as also shown very recently [62]. Similarly, e.g., SR proteins pseudo-RRMs were shown to use non-canonical RNA-binding sites [63]. 

The herein suggested motif preference of the SAFB2 RRM is found in C-rich or A-rich RNA sequences. As such, it is possibly reflected by the ENCODE-deposited C-rich 5-/6-mers derived from RBnS, and also found for RRMs earlier [51], e.g., for the IMP3 MD-RBP [52,64]. However, we find weaker interaction with the precise ENCODE RNA, and, likewise, a control RNA with a central motif, e.g., recognized by the SRSF6 RRMs [65], is bound to an equal extent. This may hint at a hidden parameter for specificity in a protein fl-context (see below). Alternatively, we suggest that poly-C or poly-A allow for an optimal stacking of ssRNA to be accommodated on the RRM surface and interaction could involve additional contacts. Thus, the central motif covered by the RNPs may be found more affine as represented by the RBnS-derived fl-directed motif. 

Current literature still leaves much space for the actual role of RNA binding by the SAFB2 RRM in a functional context. Hutter et al. find that the RRM is not needed in pre-miRNA processing by SAFB2 [23], indicating the interaction with RNA takes place via the IDRs or is mediated via interacting proteins. Nevertheless, a direct RNA-interacting role could become more evident in other SAFB2 contexts or within protein multimers, as shown for the CAC recognition by RRMs in dimeric RBPMS [51]. Of note, however, our NMR relaxation data clearly reveal the SAFB2 RRM to exist as a monomer in its isolated form. Alternatively, the SAFB RRM may be targeting RNAs in a structured context. In fact, the possibility of RBDs, specific for ssRNA, to interact with ssRNA embedded in structures has been shown for RRMs before [66] and is also discussed for KH domains [67,68]. Of a special note, our data suggest the RRM can, in principle, interact with DNA. Although highly speculative, it could, thus, in principle, fulfill moonlighting roles as an additional (facultative) DBD or even switch between the two types of nucleic acids in dependence of the cellular context and relevant mechanistic requirements. A similar assumption has earlier been made for KH domains, most recently within the helicase DDX43 [69].

We succeeded in the full backbone NMR resonance assignment of the SAFB2 SAP domain. Based on the fact that we were able to construct a high-confidence CS-Rosetta structural ensemble, this suggests an extended SAP domain of an otherwise canonical fold, conserved fold, and this is also corroborated by the unstructured N-terminal part of our starting construct (AA 21–25). We, however, find an interesting CSP pattern with respect to prominent contacts in the N-terminal helical extension, as supported by our mutational data. Extended SAP domains with N-terminal helical extensions have been found earlier (see, e.g., our DALI-derived examples used for alignments), and even appear to comprise a full third helix in their amino-terminus [50]. However, involvement of this extension in DNA binding has remained unresolved or not even discussed for the majority of them. The crystal structure of the T4 endonuclease in complex with a Holliday junction dsDNA shows the analogous N-terminal extension of its SAP domain facing the DNA interface [70], and this domain is a close structural relative of the SAFB2 SAP. Despite a large number of available high-resolution structures of SAP domains, both isolated and in larger multi-domain contexts deposited in the PDB [71], complexes with DNA have remained the exceptions or do not reveal the SAP domain itself in direct contact with DNA. Interestingly, a very recent study shows that C-terminal basic residues of the SDE2 SAP contribute to DNA-recognition [39]. Our NMR data unambiguously prove the SAFB2 SAP domain to specifically interact with dsDNA, while ssDNA is not recognized, despite a very similar fold compared to the solution structure of SDE2 SAP (Appendix A) [39]. The SAP’s apparent micromolar affinity for dsDNA is in line with other conserved dsDNA-binding domains, such as, e.g., the AT-rich interacting domain (ARID) in *Arid* proteins [72], but may be higher in full-length protein context (see below). 

Interestingly, the SAFB2 SAP domain (21–70) only shares 36/57% of sequence identity/similarity with the SAP domain of SAFA. Nonetheless, the available NMR structure (PDB 1ZRJ, unpublished) well represents the overall fold of the sparse data-derived model of SAFB2 SAP in this work (root-mean-square-distance, rmsd, 1.1 Å between Val35 and Val64) and, moreover, contains a likewise oriented N-terminal helical extension. This not only suggests a highly conserved fold of extended SAP types, but also a possible evolutionary relation between the two SAF subfamilies. Of note, however, the SAFA SAP does not contain a serine at its analogous position 31, indicating a role of this residue for DNA sequence specificity.

The most surprising finding here is the existence of the SAFB2 SAP domain intra- and inter-domain dynamics revealed by NMR. To our knowledge, this has not specifically been addressed for available SAP domain structures. We observe chemical exchange, possibly caused by movements of the two HL helices with respect to each other, that may keep the domain flexible towards the recognition of correct DNA sequences. The domain could, thus, adopt to non-canonical DNA structure or adjust to modified groove widths. Inter-helical contacts may, thus, be transient or facultative, while a set of defined interactions are needed for structural integrity. As such, the crucial residue Arg45 may be involved in salt bridges, which will explain SAP domain unfolding after mutation.

Likewise, transient conditional SAP dimers will have interesting functional consequences for SAFB2. First, a second dimerization site, apart from the previously described CC [16,23], raises the question of the physiologically present oligomer state(s) of SAFB2 (Figure 8). In fact, multiple dimerization sites would facilitate meshes, which could serve an architectural basis in SAFB2-contained granule formation and provide a complex surface pattern for specifically 3D-arranged RNAs and DNAs. Second, dimeric SAP allows for a significantly higher specificity for interaction with dsDNA stretches. The latter could be represented by a tandem motif, a palindromic sequence or looped DNA. Additionally, full SAP domain dimerization may be initiated by a cognate tandem DNA motif. Certainly, SAFB2 oligomers will be able to engage with nucleic acids at higher affinities (avidity-driven) than the low-to-medium µM-range we find for individual domains and their target motifs in this study. Full experimental structures of SAP with DNA will reveal functionally relevant pictures of SAFB2 at the chromatin.

Additional protein–protein interactions of functional impact could exist between the SAFB2 SAP and RRM domains. Indeed, RRMs have been shown to mediate protein–protein interactions earlier [73]. Clearly, we do not see any evidence for interaction based on NMR data for the exact SAFB2 domain constructs under study (Appendix A). However, we do not rule out interactions between sequence stretches more distal, respectively, which may bring the two domains in close contact in a natural context. 

The latter would yet more put into focus the question of SAFB2 simultaneously interacting with DNA and RNA. In general, such interchanging or mixed D/RNPs including SAFBs could couple the events of specific local chromatin remodeling, transcriptional regulation of miR genes and the subsequent processing of nascent transcripts, or, as shown very recently in an example, for the interplay of nuclear matrix remodeling and mRNA splicing regulation [58] by SAFB2. In this regard, a close proximity of both bindings might be favored and give rise to a functional switch, in that internal protein interactions are crucial for the formation of correct D/RNPs.

Importantly, we find comparably low affinities for both NA-binding domains of SAFB2 in our study, which, in addition, only make up a small part of the 953-residue sequence. In analogy, however, to a wide range of multi-domain NA-binding proteins [74,75], e.g., well-represented by the IGF2BP family [68], we expect that the contribution of multiple domains—including IDRs—and oligomers will sum up to modulate affinity and specificity of full-length SAFB2. In fact, this is also suggested by the available SAFB2 RBnS data in the ENCODE database that have been obtained with nM-concentrations of protein indicating the approximate K*_D_*, in this case for RNA (ENCSR558RBK). Likely, a similar increase in affinity will be given for DNA compared to this single-domain study.

SAFBs may represent yet another example of integrative protein functions as also known for other (e.g., immune-modulatory) DRBPs that act on multiple levels of gene regulation (summarized in [2]). SAFBs, though, may have very distinct functions, and some of those will relate to its capability of simultaneous NA binding in respective cellular contexts. Performing NA-binding tasks might also be steered by protein co-factors that direct SAFBs to the correct sites of action in cells. Similarly, lncRNAs could act in an architectural manner to locate SAFBs to DNA-related tasks. In fact, lncRNA binding has been shown for SAFBs in the most recent past [20,76]. 

Interestingly, binding of SAFBs has very recently been brought in correlation with the nuclear localization of linear, mostly unspliced RNAs [77], indicating an anchoring role for them in transcript processing, which is susceptible to stress in a reversible manner [58].

Based on our initial structural data, we envision that future work with distinct derivable questions towards the role of SAP and RRM and their oligomeric states will stepwise unravel the SAFB2-driven cellular mechanisms in more detail and at the atomic level.

## 4. Materials and Methods

### 4.1. Construct Design

Amino acid sequences of the human SAFB2 SAP and RRM domains were taken from UniProt [30] entry Q14151. SAP domain boundaries were initially designed to comprise aa 21 to 70, in analogy to human MKL/myocardin-like protein 1 SAP domain (PDB 2KVU, unpublished). RRM domain boundaries were designed to comprise aa 404 to 485, in analogy to the human RBM5 RRM domain (PDB 2LKZ, [43]). For both domains, *E. coli* codon-optimized DNA constructs (Eurofins Genomics) were cloned into pET21-based vector pET-Trx1a, containing an N-terminal His_6_-tag, a Thioredoxin-tag (Trx) and a TEV cleavage site via restriction digestion (*Nco*I/*Xho*I, New England BioLabs^®^) and subsequent ligation (T4 DNA Ligase, New England BioLabs^®^). For SAP, a 5.87 kDa protein was obtained theoretically upon TEV-cleavage, containing three artificial N-terminal residues (GAM), before the start of the native protein sequence at G21 of full-length SAFB2. For RRM, a 9.13 kDa protein was obtained upon TEV-cleavage, containing three artificial N-terminal residues (GAM) before the start of the native protein sequence at G404 of full-length SAFB2. 

SAP domain point mutations (S31A, R45A and K53A/S54A) were either cloned by site-directed mutagenesis (SDM) SPRINP [78] or according to NEBaseChanger^®^ SDM following the manufacturer’s protocol. The oligonucleotides used for SDM are listed in Appendix A. Like the WT SAP protein, the mutant sequences were preceded by the three artificial residues GAM.

An N-terminally truncated SAP domain (26–70) was cloned based on pET-Trx1a SAP 21–70 plasmid using oligonucleotides listed in Appendix A, according to NEBaseChanger^®^ following the manufacturer’s protocol. For SAP_26–70_, a 5.15 kDa protein was theoretically obtained upon TEV-cleavage, with a direct start of the native protein sequence at G26 of full-length SAFB2. 

### 4.2. Protein Production

Uniformly (^13^C),^15^N-labelled SAP and RRM were expressed in *E. coli* strain BL21 (DE3) in M9 minimal medium containing (2 g/L ^13^C_6_-D-glucose (Eurisotop)), 1 g/L ^15^NH_4_Cl (Cambridge Isotope Laboratories) and 50 µg/mL kanamycin. Protein expression was induced with 1 mM IPTG at OD_600_ 0.6 to 0.8 for 18 to 21 h at 18 °C. Cell pellets were resuspended in 50 mM Tris, pH 8, 300 mM NaCl and 10 mM imidazole, supplemented with 100 µL protease inhibitor mix (SERVA) and 100 µg DNase per 1 L culture. RRM buffer was additionally supplemented with 3 mM ß-mercaptoethanol. Cells were disrupted by sonication. Supernatants were cleared by centrifugation (40 min, 9000× *g*, 4 °C). The cleared supernatants were passed over a Ni^2+^-NTA gravity flow column (Sigma-Aldrich) and the His_6_-Trx-tag was cleaved overnight at 4 °C with 0.5 mg of TEV protease per 1 L of culture, while dialyzing into fresh buffer 50 mM Tris, pH 8, 300 mM NaCl (and 3 mM ß-mercaptoethanol for RRM). With a second Ni^2+^-NTA gravity flow column, TEV protease and the cleaved tag were removed. The SAFB2 SAP and RRM domains were further purified via size exclusion on a HiLoad 16/600 SD 75 (GE Healthcare), respectively. SAP size exclusion was carried out in 25 mM sodium phosphate, 150 mM NaCl, 0.02% NaN_3_, pH 6.5. Pure SAP-containing fractions were determined by SDS-PAGE. According to its retention volume in the size exclusion chromatography, the 5.87 kDa-protein is a dimer in solution. Based on calibration, the peak position of SAP corresponded to an approximate size of 11 kDa, which is in good agreement with the theoretical molecular mass of the dimeric protein (11.74 kDa). The SEC fractions of interest were pooled and concentrated using Amicon^®^ centrifugal concentrators (molecular weight cutoff 3 kDa). NMR samples were prepared in 25 mM sodium phosphate, 150 mM NaCl, 0.02% NaN_3_, pH 6.5 with 5% (*v*/*v*) D_2_O. 

RRM size exclusion was carried out in 20 mM Bis-Tris, 1 M NaCl, 2 mM TCEP, 0.02% NaN3, pH 6.5. Pure RRM-containing fractions were determined by SDS-PAGE and accordingly pooled and concentrated using Amicon^®^ centrifugal concentrators (molecular weight cutoff 3 kDa). NMR samples were prepared in 20 mM Bis-Tris, 150 mM NaCl, 2 mM TCEP, 0.02% NaN_3_, pH 6.5 with 5% (*v*/*v*) D_2_O.

### 4.3. RNA Preparation

RNA 6-mers and 7-mers (Table 1) were obtained from Dharmacon Horizon in 2 µmol quantities, deprotected and desalted. According to the yields, each RNA was dissolved in MQ-H_2_O to reach a final concentration of 2 mM. RNA was transferred to Pur-A-Lyzer Midi 1000 tube (Sigma Aldrich, St. Louis, MI, USA) and dialyzed twice against a volume of 1 L H_2_O. Concentration was determined with a NanoDrop (Thermo Fischer, Waltham, MA, USA) subsequent to NMR titration experiments.

The Bulge-RNA was produced by in-house optimized in vitro transcription and purified as described previously [53,79]. Final RNA samples were buffer-exchanged to either SAP or RRM NMR buffer conditions and homogeneity and stability monitored by denaturing PAGE.

### 4.4. NMR Data Acquisition and Analysis

All software applications related to structural biology, including structure visualization and image creation by PyMol (Schrödinger), were run via the SBGrid platform [80]. NMR measurements were carried out at the Frankfurt BMRZ using Bruker spectrometers of 600–950 MHz proton Larmor frequency, equipped with cryogenic probes and using Z-axis pulsed field gradients. All SAFB2 SAP protein samples were consistently measured in buffer of 20 mM BisTris, 150 mM NaCl, pH 6.5 including 5% of D_2_O at 298K. For the SAP backbone assignment (BMRB entry: 51700), we recorded the following experiments: HNcaCO, HNCO, HNCACB, HNcoCACB, HBHANH, HBHAcoNH and ^15^N-NOESY. The RRM was measured in an identical buffer but including 2 mM of TCEP. The initial backbone assignment was performed at pH 5.0 (BMRB entry: 51701) using HNcaCO, HNCO, HNCACB and ^15^N-NOESY experiments and amide assignments transferred to pH 6.5 via pH titration and an additional ^15^N-NOESY experiment (BMRB entry: 51724). Data acquisition and processing was undertaken using Topspin versions 3 and 4. Cosine-squared window functions were applied for apodization in all dimensions. Spectra were referenced with respect to added DSS and for ^13^C/^15^N, as suggested in [81].

^1^H-^15^N HSQC experiments for the SAP domain were acquired with a nitrogen offset at 118 ppm and a constant spectral width of 26 ppm using 96–128 indirect complex points. For the RRM domain, we used a nitrogen offset at 116 ppm and a constant spectral width of 30 ppm using 112–128 indirect complex points.

For nucleic acid titrations to the SAFB2 SAP domain, we used 96 indirect points and complex forward linear prediction until 144 points. For SAFB2 RRM titrations, 112 indirect points were recorded with complex forward linear prediction until 168 points. A reference HSQC was recorded prior to measuring the final titration point. Subsequently, apo and end-point samples were re-mixed to obtain distinct stoichiometric ratios. Combined ^1^H/^15^N chemical shift perturbations were calculated using the formula:(1)$$CSP=\sqrt{{\left(\frac{\delta N}{5}\right)}^{2}+{\left(\delta H\right)}^{2}}$$

The {^1^H}^15^N heteronuclear NOE experiments were performed as interleaved HSQC-based pseudo-3D versions including solvent suppression by WATERGATE sequence [82] and a saturation delay of 6 s from samples of 1 mM (SAP) and 600 μM (RRM) concentration. Analysis of CSPs and hetNOE ratios were performed using the CCPNMR analysis 2.4 software suite [83] and the program Sparky [84]. 

For ^15^N-relaxation data of the SAFB2 SAP domain, we used the following T1 delays: 20, 40, 80, 100, 200, 400, 600, 1000 and 1400 ms. T2 delays were 16.96, 33.92, 67.84, 101.76, 135.68, 169.6, 203.52, 271.36 and 339.2 ms. T1rho delays were 4, 8, 12, 24, 48, 72, 96, 120, 144 ms. All data were acquired with implemented temperature compensation at a field strength of 700 MHz proton Larmor frequency and 298K. Relaxation Rates R1, R2 and R1rho were derived as reciprocals of either T1, T2 and T1rho. For the SAFB2 RRM domain, relaxation data were recorded at 600 MHz proton Larmor frequency and 298 K. The following delays were used for T1: 10, 50, 150, 250, 350, 500, 750, 1500 and 2500 ms and for T2: 33.92, 50.88, 67.84, 84.8, 101.76, 118.72, 135.68, 152.64, 169.6, 203.52 and 271.36 ms. TauC values were calculated according to the formula: (2)$$TauC\;\left(ns\right)=\left(\frac{1}{2*\pi *15N*10^{9}}\right)\sqrt{\left(\frac{3}{2}\right)*\left(\frac{R2}{R1}\right)-\left(\frac{7}{6}\right)}*10^{9}$$

∗^15^N referring to the ^15^N Larmor frequency. 

CS-based structural models were calculated using the CS-Rosetta [85] web server version 3.8 (version 3.3 of the CS-Rosetta toolbox) by generating 3000 structures and no automated trimming of flexible tails. No additional restraints other than chemical shifts as deposited for the two domains and TALOS-derived backbone angles have been added. For both domains, the auto-validation returned a converged run, indicating a meaningful structural ensemble. Statistics of both runs’ ensembles of best 10 structures are shown in Appendix A. The full output is publicly available (see Data availability statement).

### 4.5. Analytical Size-Exclusion Chromatography (aSEC) and aSEC-Based MW Determination 

A Superdex75 Increase 10/300GL column was calibrated at 4 °C by loading 100 uL of premixed commercial low molecular weight standards (Cytiva) containing a mixture of five proteins in SAP buffer and running them at 0.6 mL/min using a Bio-Rad NGC FPLC. A standard calibration curve was generated from a linear fit of the partition coefficient (Kav) versus the log molecular weight of the protein standards. The apparent molecular weight of SAP wild-type and the mutants was calculated by interpolation from their Kav using the equation derived from the linear fit. Analytical SEC runs at 4 °C were performed by loading 100 µL of protein samples (~4 mg/mL) onto a Superdex75 Increase 10/300GL equilibrated with SAP buffer using a Bio-Rad NGC FPLC at 0.6 mL/min. The run was monitored with UV absorbance at 215 nm. All aSEC runs were analyzed using ChromLab_v6 (Bio-Rad) and the traces plotted in OriginPro (see Appendix A).

### 4.6. Size Exclusion Chromatography Coupled with Multi-Angle Light Scattering (SEC-MALS)

SEC-MALS was performed at 4 °C using a Superdex75 Increase 10/300GL column, a light-scattering detector (TREOS) and refractometer (Optilab rEX) from Wyatt Technology and a UV detector, HPLC pump and degasser from Jasco. System was equilibrated with 3 column volumes of SAP buffer. All buffers were filtered through 0.1 µm pore size Durapore PVDF membrane filters (Merck) following a recirculation through the system for 20 h at 0.5 mL/min to improve the baseline by removing air bubbles and particles by degasser and pre-injection filter (0.1 µm). Per measurement, 270 µg of protein in 100 µL buffer were injected and analyzed at a flow rate of 0.5 mL/min. Light scattering detector was calibrated by using monomeric BSA (Merck). The obtained signals were processed with the ASTRA software package version 5.3.4.13 (Wyatt Technology, Haverhill, UK). The concentration was determined by the refractometer using a refractive index increment dn/dc at 20.5 °C and 658 nm as calculated by SEDFIT v16.1c [86].

### 4.7. Electromobility Shifts Assays (EMSAs) 

EMSAs for the SAP domain were performed with DNA such that the single strand or one of the two strands in dsDNA was 5′-labeled with fluorescein (6-FAM, Sigma-Aldrich) (Table 1). For each reaction, 0.1 pmol of either single- or double-stranded DNA were mixed with 0,6 µg of yeast tRNA (Roche), 10 mM MgCl_2_ and SAP-buffer (25 mM sodium phosphate, 150 mM NaCl, 0.02% NaN_3_, pH 6.5)/protein in different volumes/concentrations to match a final reaction volume of 20 µL. Additionally, 3 µL of loading buffer were added immediately prior to loading the samples (10 µL of each reaction) onto a 6% polyacrylamide gel. Gel-electrophoresis was run for 50 min at 80 V. Finally, the gels were imaged inside the glass plates with a Typhoon Imager (GE Healthcare) with a laser at 488 nm excitation and an emission filter at 520 nm.

## Figures and Tables

**Figure 1 ijms-24-03286-f001:**
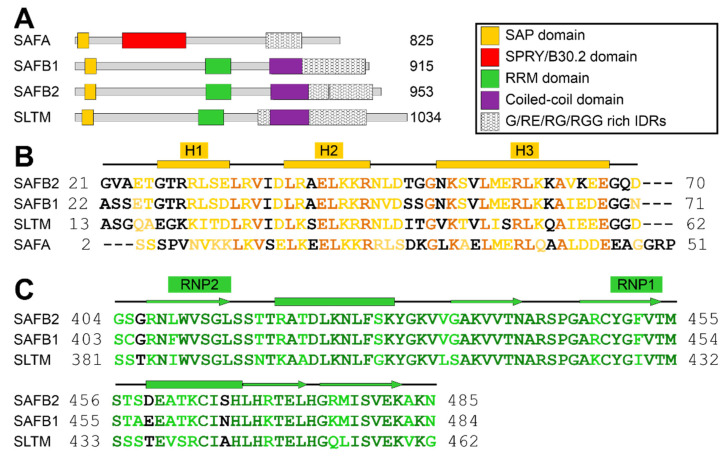
Scaffold attachment factor proteins (SAF) type A (hnRNP U) and B (SAFB1, SAFB2 and SLTM). (**A**) Domain organizations of human SAFA, SAFB1, SAFB2 and SLTM as annotated in Uniprot [30] (IDs: Q00839, Q15424, Q14151, Q9NWH9). (**B**) Sequence alignment of the SAP domain (based on domain boundaries of the SAFB2 SAP defined in this study) with residues color-coded by conservation amongst the four proteins according to Clustal Omega [31] (black—no conservation, light orange—weak-to-strong conservation, dark orange—full conservation). Above, secondary structure elements derived from the SCS analysis for SAFB2 SAP in Figure 2 are given, with rectangles indicating α-helices. The helix–loop–helix DNA-binding motif spans helices 2 (H2) and 3 (H3). (**C**) Sequence alignment of the SAFB RRM domains (boundaries are based on those of the SAFB2 RRM domain defined in this study) with residues colored according to the degree of conservation as by Clustal Omega (black—no conservation, light green—weak-to-strong conservation, dark green—full conservation). Above, the two conserved RNA-binding motifs (RNP) and secondary structure elements derived from the SCS analysis for SAFB2 RRM in Figure 3 are given, with rectangles indicating α-helices and arrows indicating β-strands, respectively.

**Figure 2 ijms-24-03286-f002:**
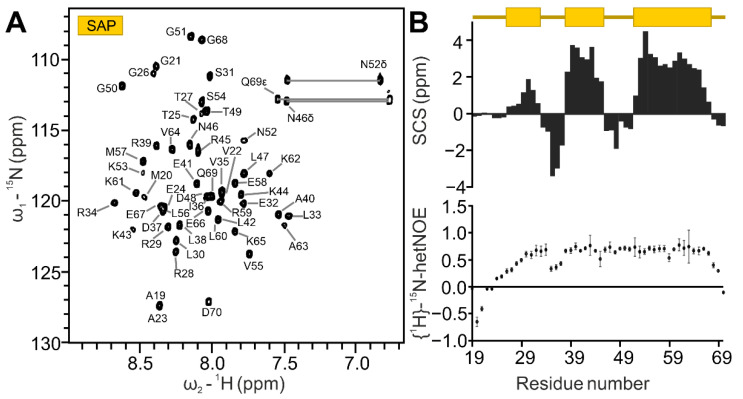
Resonance assignment of the SAFB2 SAP domain. (**A**) ^1^H-^15^N HSQC of the SAP domain amide groups. Straight lines indicate side chain assignments. (**B**) Combined Cα/Cβ carbon secondary chemical shifts (SCS) (upper panel, as suggested by [32]—rectangles indicating α-helices) and the {^1^H}^15^N heteronuclear NOE values (lower panel) of SAFB2 SAP plotted against the residue number. Errors are calculated from the program CCPNMR Analysis.

**Figure 3 ijms-24-03286-f003:**
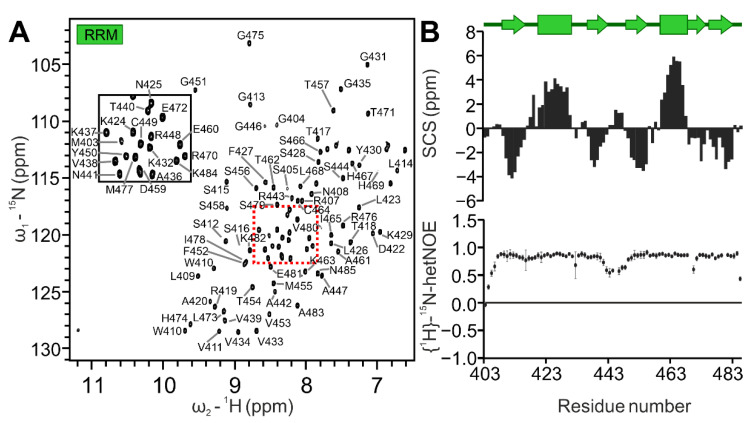
Resonance assignment of the SAFB2 RRM domain. (**A**) ^1^H-^15^N HSQC of the RRM domain amide groups. Assignments of peaks in the red dashed-lined box are shown in the inset zoom-in. See also Appendix A. (**B**) Combined Cα/Cβ carbon secondary chemical shifts (SCS) (upper panel, as suggested by [32]—rectangles indicating α helices, arrows indicating β-strands) and the {^1^H}^15^N heteronuclear NOE values (lower panel) of the SAFB2 RRM plotted against residue numbers. Errors are calculated from the program CCPNMR Analysis.

**Figure 4 ijms-24-03286-f004:**
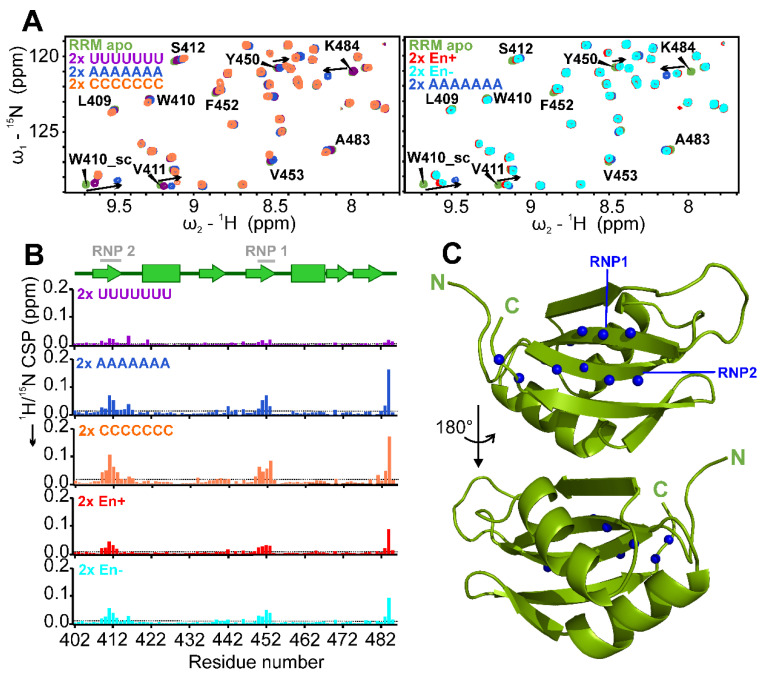
RNA-binding preferences of the SAFB2 RRM domain. (**A**) ^1^H-^15^N HSQC zoom-ins showing overlays of 2× molar excess of RNAs as indicated (color code depicted in upper left corner) with the apo RRM, respectively. Assignments for RNP residues involved in RNA interaction are included. Addition of G_7_ resulted in severe precipitation and was, thus, not evaluated accordingly. See Appendix A for full spectra. (**B**) CSP plots of individual RNAs over amino acid sequence. Indicated above is the RRM secondary structure with RNP1 and RNP2 highlighted. Dotted grey lines represent the CSP average. (**C**) Mapping of the ten significantly perturbed CS for residues shown as spheres (N-atoms) to a structural model of the SAFB2 RRM domain, i.e., the lowest energy representative of a CS-Rosetta-derived ensemble (Appendix A). Maximum CSPs locate to the two RNP motifs and the C-terminal tail, respectively.

**Figure 5 ijms-24-03286-f005:**
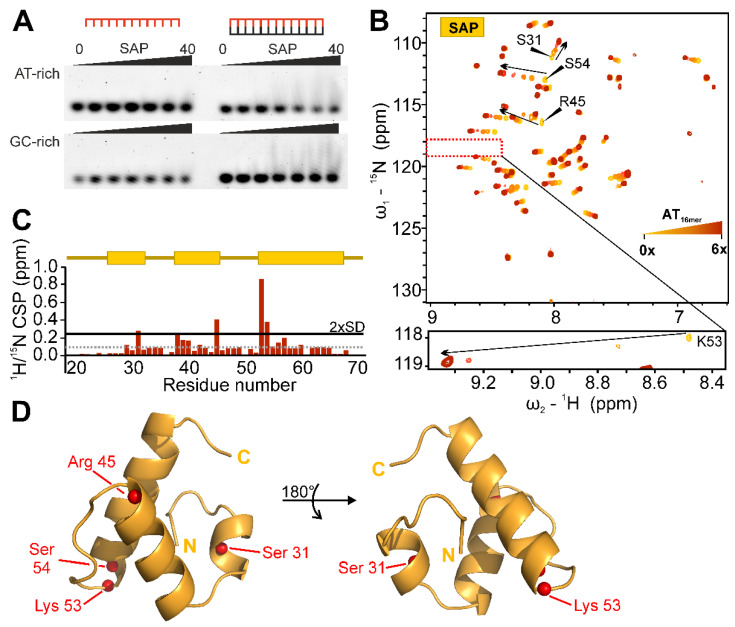
DNA-binding preferences of the SAFB2 SAP domain. (**A**) EMSAs of SAP domain with ss (**left**) and ds (**right**) DNAs, either AT-rich (**top**) or GC-rich (**bottom**) (with *sense* sequences as follows: AT_13mer_ 5′-GCAATAAATACG-3′; GC_13mer_ 5′-GCCCCCGCGCCCG-3′). Increasing SAP protein concentration (0–40 µM) is indicated by black bar above gel pictures. EMSAs shown here are representatives of technical triplicates (Appendix A). (**B**) ^1^H-^15^N HSQC of SAP titrated with increasing molar ratios (0.5×, 1×, 1.85×, 4×, 6×) of the dsDNA AT_16mer_ (5′-GCGCACAATATAACGC-3′). The most perturbed resonances are labelled with their assignments. The zoom-in of the boxed region below shows the highly conserved Lys53 (see Figure 1), which is most strongly affected. Note the different contour levels used to visualize this peak relative to the settings of the full spectrum above. (**C**) CSP plots of 6× molar excess of AT_16mer_ plotted over amino acid sequence. Indicated above is the SAP secondary structure. Dotted grey line represents the CSP average, the solid black line the threshold of 2× standard deviation. (**D**) Mapping of the four significantly perturbed CS for residues shown as spheres (N-atoms) plotted on a structural model of the SAFB2 SAP domain. The structural model is the lowest energy representative of a CS-Rosetta-derived ensemble.

**Figure 6 ijms-24-03286-f006:**
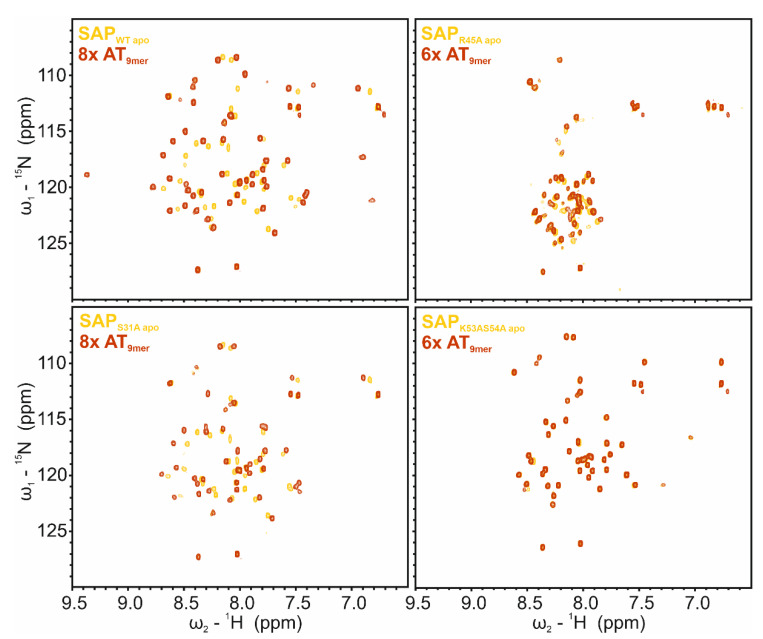
DNA-binding capacity of SAFB2 SAP domain mutants. ^1^H-^15^N HSQC overlay of apo SAP proteins (yellow) with 6- or 8-fold excess (red) of AT_9mer_ (5′-CAATATAAC-3′), as indicated. Top panel: SAP_WT_ (**left**) and SAP_R45A_ (**right**); bottom panel: SAP_S31A_ (**left**) and SAP_K53A/S54A_ (**right**). Contour levels are scaled individually between proteins, for peaks to be visible. Contour levels for each apo and DNA-bound spectra of one protein are adjusted.

**Figure 7 ijms-24-03286-f007:**
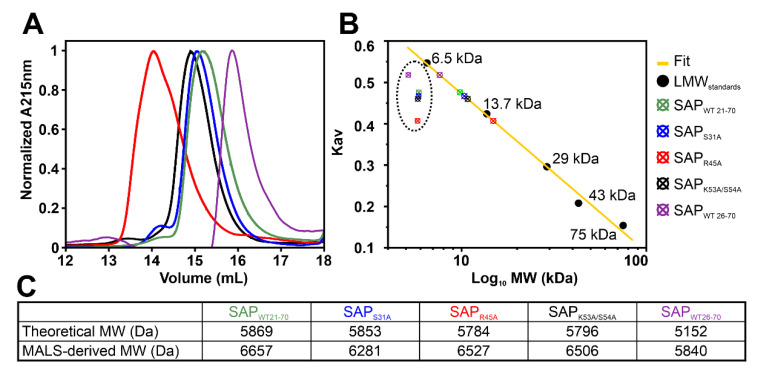
SAFB2 SAP domain mutants. (**A**) aSEC of SAP wild-type domain and mutants. (**B**) Estimation of molecular weights (MW) based on MW standards. Shown is the partition coefficient (Kav) plotted as a function of MW. Colored points (in dotted circle) are the Kav from aSEC over their theoretical MWs (expecting a monomer), respectively. Colored points on the fit line are the MWs determined based on the calibration standards (see also Appendix A). (**C**) SEC-MALS-derived MWs of the SAP wild-type domain and mutants compared to the theoretical MW of a mono-mer.

**Figure 8 ijms-24-03286-f008:**
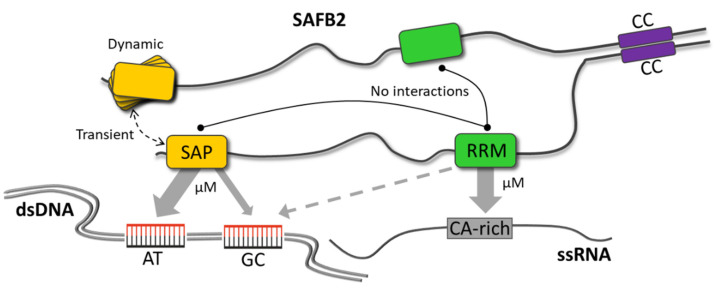
Summary of nucleic acid-binding preferences of SAFB2 SAP and RRM domains as determined in this study. In addition, possible inter- and intra-molecular domain interactions based on current experimental knowledge are depicted. SAFB2 and its domains are not shown in scale.

**Table 1 ijms-24-03286-t001:** Nucleic acid sequences used to examine the binding preferences of the SAFB2 RRM and SAP domains in this study.

Name	Sequence 5′ → 3′	Usage
DNA Oligos used for binding studies		
AT_9mer_fw_	CAATATAAC	Hybridized to AT_9mer_, for NMR titration.
AT_9mer_rev_	GTTATATTG	
AT_16mer_fw_	GCGCACAATATAACGC	Hybridized to AT_16mer_, for NMR titration.
AT_16mer_rev_	GCGTTATATTGTGCGC	
GC_16mer_fw_	CGCCCCCGCGCCCGCG	Hybridized to GC_16mer_, for NMR titration.
GC_16mer_rev_	CGCGGGCGCGGGGGCG	
AT_13mer_fw_	[FAM]-GCAATAAATACG	Hybridized to AT_13mer_, for fluorescent EMSAs.
AT_13mer_rev_	CGTATTTATTGC	
GC_13mer_fw_	[FAM]-GCCCCCGCGCCCG	Hybridized to GC_13mer_, for fluorescent EMSAs.
GC_13mer_rev_	CGGGCGCGGGGGC	
RNAs used for binding studies		
En+	UACACC	ENCODE ^(a)^ consensus
En−	CGGACU	ENCODE ^(a)^ control
U_7_	UUUUUUU	Poly-N U-rich
A_7_	AAAAAAA	Poly-N A-rich
C_7_	CCCCCCC	Poly-N C-rich
G_7_	GGGGGGG	Poly-N G-rich
Bulge	GGAUUUUAUGGGGCACGGACAACCCAUAUCCUGAU	Multi-characteristic model RNA

^(a)^ Motif based on an RBnS-derived entry available in ENCODE [33], identifier ENCSR558RBK.

## Data Availability

Full results of CS-Rosetta runs for SAP and RRM domains are accessible under entries https://csrosetta.bmrb.io/entry/36b19517843f and https://csrosetta.bmrb.io/entry/54b3a1e3a6cb, respectively, accessed on 27 September 2022. Backbone chemical shifts of SAFB2 SAP and RRM have been deposited in the BMRB with IDs 51700, 51701 and 51724, respectively.

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
