# Peer review of "Insight into the Structural Basis for Dual Nucleic Acid—Recognition by the Scaffold Attachment Factor B2 Protein"

_ijms, 2023, doi:10.3390/ijms24043286_

Round 1
Reviewer 1 Report
The main query of the authors is to find out the details binding of DNA and RNA sequences on specific domain of SAFB protein. They also search the specificity of the DNA and RNA binding activates. The authors used theoretical MD simulation and Experimentally NMR techniques which is highly sophisticate and modern techniques. As per my opinion the techniques are sufficient. Though author my use atomic force microscopy to clearly observe the binding domain. In present day the binding of any molecule to a specific domain and macromolecule are topic of big challenge. Two D NMR is a modern technique to identify the interaction of small
molecules in atomic level.
The results and conclusion are good and reference are appropriate. Author did very good work.
Can accepted for publication.
Author Response
We thank the reviewer for the positive feedback and overall appreciation of our work. We have included the information of SAFB2’s full-length size relative to the small NA-binding domains as a clear information for the reader in the text. As detailed below, we have addressed the remaining, justified comments.
Reviewer 2 Report
In the manuscript “Insight into the structural basis for dual nucleic acid - recognition by the Scaffold Attachment Factor B2 protein” by Korn et al., the authors tried to use structural approaches to study the mechanism behind the DNA- and RNA-binding functions of SAFB2. They used NMR to explore two small fragments containing the DNA-binding SAP domain (5.87 kDa) and the RNA-binding RPM domain (9.13 kDa). However, it is noteworthy to point out that full-length SAFB2 has 953 amino acids. Additionally, the two fragments used in this study has shown very low affinity in DNA binding.
Here are the comments:
1. Supplementary Fig. 1, the gel has shown too much contrast.
2. On page 8, Figure 5A, the EMSA shows very poor binding of SAP domains towards ssDNA or dsDNA substrates. The authors should discuss whether this low activity is caused by short fragments used or not. What is the binding activity of the full-length protein?
3. On page 9, Figure 6, the authors analyzed that mutants’ abilities to bind dsDNA using NMR. EMSA should also be used to confirm the results as an independent orthogonal test.
Author Response
In the manuscript “Insight into the structural basis for dual nucleic acid - recognition by the Scaffold Attachment Factor B2 protein” by Korn et al., the authors tried to use structural approaches to study the mechanism behind the DNA- and RNA-binding functions of SAFB2. They used NMR to explore two small fragments containing the DNA-binding SAP domain (5.87 kDa) and the RNA-binding RPM domain (9.13 kDa). However, it is noteworthy to point out that full-length SAFB2 has 953 amino acids. Additionally, the two fragments used in this study has shown very low affinity in DNA binding.
We thank the reviewer for the positive feedback and overall appreciation of our work. We have included the information of SAFB2’s full-length size relative to the small NA-binding domains as a clear information for the reader in the text. As detailed below, we have addressed the remaining, justified comments.
Here are the comments:
- Supplementary Fig. 1, the gel has shown too much contrast.
We exchanged the respective gel, showing now an original, unprocessed gel image. Note that Suppl. Fig. 1 now also contains an additional panel with NH sidechain assignments of the RRM (also deposited in the BMRB).
- On page 8, Figure 5A, the EMSA shows very poor binding of SAP domains towards ssDNA or dsDNA substrates. The authors should discuss whether this low activity is caused by short fragments used or not. What is the binding activity of the full-length protein?
We discussed the low apparent affinity of SAP with regard to full-length protein in the discussion.
The SAP domain shows apparent micromolar affinities for the DNAs used in our study. While this affinity range is at the limit of EMSA-based analysis, it is perfectly fit to be studied by NMR. NMR of the SAP domain further confirms (SCS and het-NOE plots in Figure 2B) that the chosen domain boundaries are sufficient. Similar to other multidomain nucleic acid-binding proteins (composed of folded domains as well as intrinsically disordered regions (IDRs)), it can be assumed that SAFB2’s in vivo DNA-binding affinity is achieved through the contribution of multiple domains. In particular, IDRs have been described to increase affinity and modulate specificity, e.g. through protein-protein or protein-nucleic acid interactions (see e.g. Pontoriero et al., 2022 10.3390/biom12070929 or Ottoz and Berchowitz, 2020 doi/10.1098/rsob.200328). Further, they are often drivers of phase separation and compartmentalization. An increase in affinity by three orders of magnitude for full-length proteins vs their respective single-domains is thus common (e.g. low nanomolar vs. micromolar, see Schneider et al., 2019 s41467-019-09769-8). In case of SAFBs, DNA-binding will likely be modulated by dimerization as well, and thus by effects of avidity.
We do not provide a Kd value of fl-SAFB2 for DNA or RNA, but refer to the ENCODE entry for SAFB2 RBnS showing the much higher affinity for RNA in the fl-context compared to our RRM-only data. We suggest, the same is true for DNA and have included this in the discussion now.
While it will be of huge interest to study the detailed mechanism of DNA (and RNA) encountering of SAFB2 comprehensively (in full-length context), it is a requirement to understand the individual components of the full-length protein. By applying solution NMR, we could clearly demonstrate SAP to specifically interact with dsDNA, but not ssDNA, and a preference for AT- over GC-rich DNA, which appears more complicated to reveal starting from a full-length protein. Thus, our results allow to assign SAFB2’s SAP domain to the dsDNA-binding type of SAP, different from the ssDNA-binding SAPs. This will serve as the basis for understanding the mechanism of full-length SAFB2 in cells.
- On page 9, Figure 6, the authors analyzed that mutants’ abilities to bind dsDNA using NMR. EMSA should also be used to confirm the results as an independent orthogonal test.
We agree that an independent and orthogonal experiment, supporting the NMR-derived results for SAP-mutations, is a valuable addition. We did the respective experiments and included them in the new Supplementary Figure 9. As shown and stated, they indeed well confirm the NMR results.